# Generating Diverse High-Fidelity Images with VQ-VAE-2

**Ali Razavi**[*]
DeepMind
alirazavi@google.com

**Aäron van den Oord**[*]
DeepMind
avdnoord@google.com

**Oriol Vinyals**
DeepMind
vinyals@google.com

## Abstract

We explore the use of Vector Quantized Variational AutoEncoder (VQ-VAE) models for large scale image generation. To this end, we scale and enhance the autoregressive priors used in VQ-VAE to generate synthetic samples of much higher coherence and fidelity than possible before. We use simple feed-forward encoder and decoder networks, making our model an attractive candidate for applications where the encoding and/or decoding speed is critical. Additionally, VQ-VAE requires sampling an autoregressive model only in the compressed latent space, which is an order of magnitude faster than sampling in the pixel space, especially for large images. We demonstrate that a multi-scale hierarchical organization of VQ-VAE, augmented with powerful priors over the latent codes, is able to generate samples with quality that rivals that of state of the art Generative Adversarial Networks on multifaceted datasets such as ImageNet, while not suffering from GAN's known shortcomings such as mode collapse and lack of diversity.

## 1   Introduction

Deep generative models have significantly improved in the past few years [5, 27, 25]. This is, in part, thanks to architectural innovations as well as computation advances that allows training them at larger scale in both amount of data and model size. The samples generated from these models are hard to distinguish from real data without close inspection, and their applications range from super resolution [21] to domain editing [44], artistic manipulation [36], or text-to-speech and music generation [25].

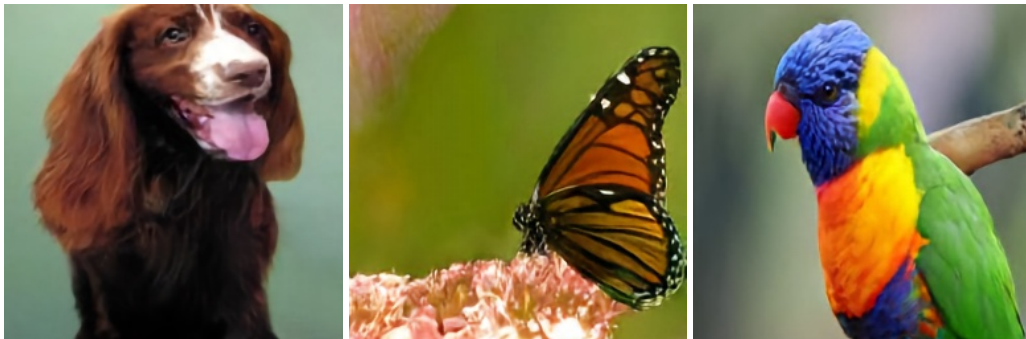

Figure 1: Class-conditional 256x256 image samples from a two-level model trained on ImageNet.

We distinguish two main types of generative models: likelihood based models, which include VAEs [16, 31], flow based [9, 30, 10, 17] and autoregressive models [20, 39]; and implicit generative

---

[*]Equal contributions.

models such as Generative Adversarial Networks (GANs) [12]. Each of these models offer several trade-offs such as sample quality, diversity, speed, etc.

GANs optimize a minimax objective with a generator neural network producing images by mapping random noise onto an image, and a discriminator defining the generators' loss function by classifying its samples as real or fake. Larger scale GAN models can now generate high-quality and high-resolution images [5, 14]. However, it is well known that samples from these models do not fully capture the diversity of the true distribution. Furthermore, GANs are challenging to evaluate, and a satisfactory generalization measure on a test set to assess overfitting does not yet exist. For model comparison and selection, researchers have used image samples or proxy measures of image quality such as Inception Score (IS) [33] and Fréchet Inception Distance (FID) [13].

In contrast, likelihood based methods optimize negative log-likelihood (NLL) of the training data. This objective allows model-comparison and measuring generalization to unseen data. Additionally, since the probability that the model assigns to *all* examples in the training set is maximized, likelihood based models, in principle, cover all modes of the data, and do not suffer from the problems of mode collapse and lack of diversity seen in GANs. In spite of these advantages, directly maximizing likelihood in the pixel space can be challenging. First, NLL in pixel space is not always a good measure of sample quality [37], and cannot be reliably used to make comparisons between different model classes. There is no intrinsic incentive for these models to focus on, for example, global structure. Some of these issues are alleviated by introducing inductive biases such as multi-scale [38, 39, 29, 22] or by modeling the dominant bit planes in an image [18, 17].

In this paper we use ideas from lossy compression to relieve the generative model from modeling negligible information. Indeed, techniques such as JPEG [43] have shown that it is often possible to remove more than 80% of the data without noticeably changing the perceived image quality. As proposed by [41], we compress images into a discrete latent space by vector-quantizing intermediate representations of an autoencoder. These representations are over 30x smaller than the original image, but still allow the decoder to reconstruct the images with little distortion. The prior over these discrete representations can be modeled with a state of the art PixelCNN [39, 40] with self-attention [42], called PixelSnail [7]. When sampling from this prior, the decoded images also exhibit the same high quality and coherence of the reconstructions (see Fig. 1). Furthermore, the training and sampling of this generative model over the discrete latent space is also 30x faster than when directly applied to the pixels, allowing us to train on much higher resolution images. Finally, the encoder and decoder used in this work retains the simplicity and speed of the original VQ-VAE, which means that the proposed method is an attractive solution for situations in which fast, low-overhead encoding and decoding of large images are required.

## 2 Background

### 2.1 Vector Quantized Variational AutoEncoder

The VQ-VAE model [41] can be better understood as a communication system. It comprises of an encoder that maps observations onto a sequence of discrete latent variables, and a decoder that reconstructs the observations from these discrete variables. Both encoder and decoder use a shared codebook. More formally, the encoder is a non-linear mapping from the input space, $x$, to a vector $E(x)$. This vector is then quantized based on its distance to the prototype vectors in the codebook $e_k, k \in 1 \ldots K$ such that each vector $E(x)$ is replaced by the index of the nearest prototype vector in the codebook, and is transmitted to the decoder (note that this process can be lossy).

$$\text{Quantize}(E(\mathbf{x})) = \mathbf{e}_k \quad \text{where } k = \arg\min_j ||E(\mathbf{x}) - \mathbf{e}_j|| \tag{1}$$

The decoder maps back the received indices to their corresponding vectors in the codebook, from which it reconstructs the data via another non-linear function. To learn these mappings, the gradient of the reconstruction error is then back-propagated through the decoder, and to the encoder using the straight-through gradient estimator. The VQ-VAE model incorporates two additional terms in its objective to align the vector space of the codebook with the output of the encoder. The *codebook loss*, which only applies to the codebook variables, brings the selected codebook $\mathbf{e}$ close to the output of the encoder, $E(\mathbf{x})$. The *commitment loss*, which only applies to the encoder weights, encourages the output of the encoder to stay close to the chosen codebook vector to prevent it from fluctuating too frequently from one code vector to another. The overall objective is described in equation 2,

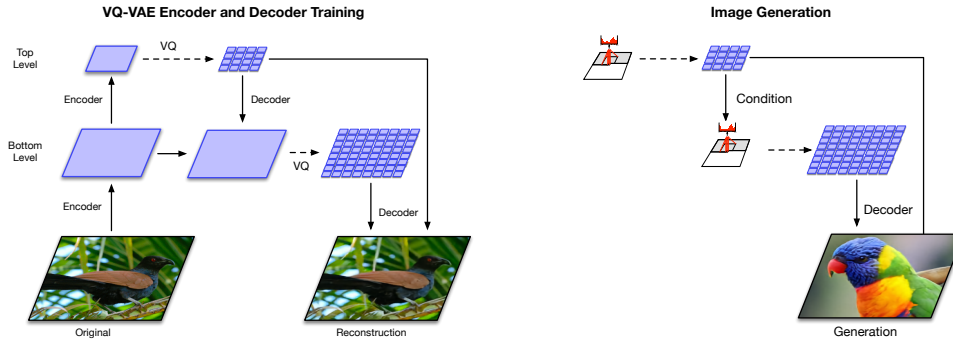

(a) Overview of the architecture of our hierarchical VQ-VAE. The encoders and decoders consist of deep neural networks. The input to the model is a $256 \times 256$ image that is compressed to quantized latent maps of size $64 \times 64$ and $32 \times 32$ for the *bottom* and *top* levels, respectively. The decoder reconstructs the image from the two latent maps.

(b) Multi-stage image generation. The top-level PixelCNN prior is conditioned on the class label, the bottom level PixelCNN is conditioned on the class label as well as the first level code. Thanks to the feed-forward decoder, the mapping between latents to pixels is fast. (The example image with a parrot is generated with this model).

Figure 2: VQ-VAE architecture.

where $\mathbf{e}$ is the quantized code for the training example $\mathbf{x}$, $E$ is the encoder function and $D$ is the decoder function. The operator $sg$ refers to a stop-gradient operation that blocks gradients from flowing into its argument, and $\beta$ is a hyperparameter which controls the reluctance to change the code corresponding to the encoder output.

$$\mathcal{L}(\mathbf{x}, D(\mathbf{e})) = ||\mathbf{x} - D(\mathbf{e})||_2^2 + ||sg[E(\mathbf{x})] - \mathbf{e}||_2^2 + \beta ||sg[\mathbf{e}] - E(\mathbf{x})||_2^2 \qquad (2)$$

As proposed in [41], we use the exponential moving average updates for the codebook, as a replacement for the codebook loss (the second loss term in Equation equation 2):

$$N_i^{(t)} := N_i^{(t-1)} * \gamma + n_i^{(t)}(1 - \gamma), \quad m_i^{(t)} := m_i^{(t-1)} * \gamma + \sum_j^{n_i^{(t)}} E(x)_{i,j}^{(t)}(1 - \gamma), \quad e_i^{(t)} := \frac{m_i^{(t)}}{N_i^{(t)}}$$

where $n_i^{(t)}$ is the number of vectors in $E(x)$ in the mini-batch that will be quantized to codebook item $e_i$, and $\gamma$ is a decay parameter with a value between 0 and 1 (default $\gamma = 0.99$ is used in all experiments). We use the released VQ-VAE implementation in the Sonnet library [2] [3].

## 3 Method

The proposed method follows a two-stage approach: first, we train a hierarchical VQ-VAE (see Fig. 2a) to encode images onto a discrete latent space, and then we fit a powerful PixelCNN prior over the discrete latent space induced by all the data.

### 3.1 Stage 1: Learning Hierarchical Latent Codes

As opposed to *vanilla* VQ-VAE, in this work we use a hierarchy of vector quantized codes to model large images. The main motivation behind this is to model local information, such as texture, separately from global information such as shape and geometry of objects. The prior model over each level can thus be tailored to capture the specific correlations that exist in that level. The structure of our multi-scale hierarchical encoder is illustrated in Fig. 2a, with a *top* latent code which models global information, and a *bottom* latent code, conditioned on the top latent, responsible for representing local details (see Fig. 3). We note if we did not condition the bottom latent on the top latent, then the top latent would need to encode every detail from the pixels. We therefore allow each level in the

hierarchy to separately depend on pixels, which encourages encoding complementary information in each latent map that can contribute to reducing the reconstruction error in the decoder.

For $256 \times 256$ images, we use a two level latent hierarchy. As depicted in Fig. 2a, the encoder network first transforms and downsamples the image by a factor of $4$ to a $64 \times 64$ representation which is quantized to our bottom level latent map. Another stack of residual blocks then further scales down the representations by a factor of two, yielding a top-level $32 \times 32$ latent map after quantization. The decoder is similarly a feed-forward network that takes as input all levels of the quantized latent hierarchy. It consists of a few residual blocks followed by a number of strided transposed convolutions to upsample the representations back to the original image size.

### 3.2 Stage 2: Learning Priors over Latent Codes

In order to further compress the image, and to be able to sample from the model learned during stage 1, we learn a prior over the latent codes. Fitting prior distributions using neural networks from training data has become common practice, as it can significantly improve the performance of latent variable models [6]. This procedure also reduces the gap between the *marginal posterior* and the prior. Thus, latent variables sampled from the learned prior at test time are close to what the decoder network has observed during training which results in more coherent outputs. From an information theoretic point of view, the process of fitting a prior to the learned posterior can be considered as lossless compression of the latent space by re-encoding the latent variables with a distribution that is a better approximation of their true distribution, and thus results in bit rates closer to Shannon's entropy. Therefore the lower the gap between the true entropy and the negative log-likelihood of the learned prior, the more realistic image samples one can expect from decoding the latent samples.

In the VQ-VAE framework, this auxiliary prior is modeled with a powerful, autoregressive neural network such as PixelCNN in a post-hoc, second stage. The prior over the top latent map is responsible for structural global information. Thus, we equip it with multi-headed self-attention layers as in [7, 26] so it can benefit from a larger receptive field to capture correlations in spatial locations that are far apart in the image. In contrast, the conditional prior model for the bottom level over latents that encode local information will operate at a larger resolution. Using self-attention layers as in the top-level prior would not be practical due to memory constraints. For this prior over local information, we thus find that using large conditioning stacks (coming from the top prior) yields good performance (see Fig. 2b). The hierarchical factorization also allows us to train larger models: we train each prior separately, thereby leveraging all the available compute and memory on hardware accelerators. Please refer to Appendix A for the details of the architecture and hyperparameters.

### 3.3 Trading off Diversity with Classifier Based Rejection Sampling

Unlike GANs, probabilistic models trained with the maximum likelihood objective are forced to model *all* of the training data distribution. This is because the MLE objective can be expressed as the forward KL-divergence between the data and model distributions, which would be driven to infinity if an example in the training data is assigned zero mass. While the coverage of all modes in the data distribution is an appealing property of these models, the task is considerably more difficult than

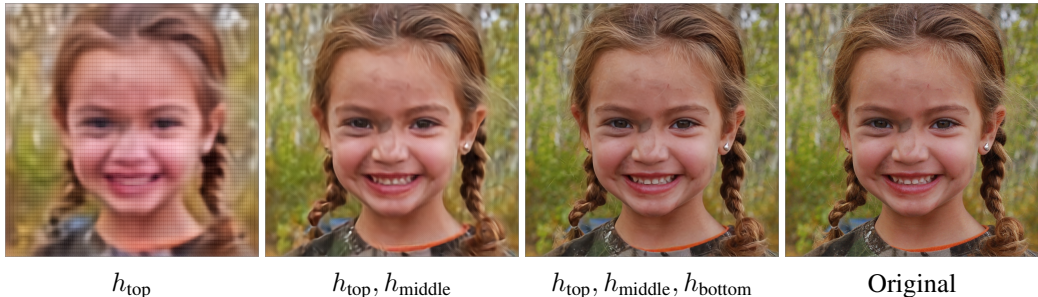

$h_{\text{top}}$        $h_{\text{top}}, h_{\text{middle}}$        $h_{\text{top}}, h_{\text{middle}}, h_{\text{bottom}}$        Original

Figure 3: Reconstructions from a hierarchical VQ-VAE with three latent maps (top, middle, bottom). The rightmost image is the original. Each latent map adds extra detail to the reconstruction. These latent maps are approximately 3072x, 768x, 192x times smaller than the original image (respectively).

adversarial modeling, since likelihood based models need to fit all the modes present in the data. Furthermore, ancestral sampling from autoregressive models can in practice induce errors that can accumulate over long sequences and result in samples with reduced quality. Recent GAN frameworks [5, 2] have proposed automated procedures for sample selection to trade-off diversity and quality. In this work, we also propose an automated method for trading off diversity and quality of samples based on the intuition that the closer our samples are to the true data manifold, the more likely they are classified to the correct class labels by a pre-trained classifier. Specifically, we use a classifier network that is trained on ImageNet to score samples from our model according to the probability the classifier assigns to the correct class. Note that we only use this classifier for the *quantitive* metrics in this paper (such as FID, IS, Precision, Recall) to trade off diversity with quality. None of the samples in this manuscript are sampled using this classifier (please follow the link in the Appendix Section to see these).

## 4    Related Works

The foundation of our work is the VQ-VAE framework of [41]. Our prior network is based on Gated PixelCNN [40] augmented with self-attention [42], as proposed in [7].

BigGAN [5] is currently state-of-the-art in FID and Inception scores, and produces high quality high-resolution images. The improvements in BigGAN come mostly from incorporating architectural advances such as self-attention, better stabilization methods, scaling up the model on TPUs and a mechanism to trade-off sample diversity with sample quality. In our work we also investigate how the addition of some of these elements, in particular self-attention and compute scale, indeed also improve the quality of samples of VQ-VAE models. Recent work has also been proposed to generate high resolution images with likelihood based models include Subscale Pixel Networks of [22]. Similar to the parallel multi-scale model of [29], SPN imposes a partitioning on the spatial dimensions, but unlike [29], SPN does not make the corresponding independence assumptions, whereby it trades sampling speed with density estimation performance and sample quality.

Hierarchical latent variables have been proposed in e.g. [31]. Specifically for VQ-VAE, [8] uses a hierarchy of latent codes for modeling and generating music using a WaveNet decoder. The specifics of the encoding is however different from ours: in our work, the bottom levels of hierarchy do not exclusively refine the information encoded by the top level, but they extract complementary information at each level, as discussed in Sect. 3.1. Additionally, as we are using simple, feed-forward decoders and optimizing mean squared error in the pixels, our model does not suffer from, and thus needs no mitigation for, the hierarchy collapse problems detailed in [8]. Concurrent to our work, [11] extends [8] for generating high-resolution images. The primary difference to our work is the use of autoregressive decoders in the pixel space. In contrast, for reasons detailed in Sect. 3, we use autoregressive models exclusively as priors in the compressed latent space, which simplifies the model and greatly improves sampling speed. Additionally, the same differences with [8] outlined above also exist between our method and [11].

Improving sample quality by rejection sampling has been previously explored for GANs [2] as well as for VAEs [4] which combines a learned rejecting sampling proposal with the prior in order to reduce its gap with the aggregate posterior. Neural networks have recently been used towards *learned* image compression. For lossy image compression, [24] trains hierarchical and autoregressive priors jointly to improve the entropy coding part of the compression system. L3C [23] is a parallel architecture proposed for lossless image compression that uses jointly learned with auxiliary latent spaces to achieve speedups in sampling compared to autoregressive models. Using GANs for extremely low rate compression is explored in [35] and [1].

## 5    Experiments

Objective evaluation and comparison of generative models, specially across model families, remains a challenge [37]. Current image generation models trade-off sample quality and diversity (or precision vs recall [32]). In this section, we present quantitative and qualitative results of our model trained on ImageNet $256 \times 256$. Sample quality is indeed high and sharp, across several representative classes as can be seen in the class conditional samples provided in Fig. 5. In terms of diversity, we provide samples from our model juxtaposed with those of BigGAN-deep [5], the state of the art GAN model

|  | Train NLL | Validation NLL | Train MSE | Validation MSE |
|---|---|---|---|---|
| Top prior | 3.40 | 3.41 | - | - |
| Bottom prior | 3.45 | 3.45 | - | - |
| VQ Decoder | - | - | 0.0047 | 0.0050 |

Table 1: Train and validation negative log-likelihood (NLL) for top and bottom prior measured by encoding train and validation set resp., as well as Mean Squared Error for train and validation set. The small difference in both NLL and MSE suggests that neither the prior network nor the VQ-VAE overfit.

[4] in Fig. 5. As can be seen in these side-by-side comparisons, VQ-VAE is able to provide samples of comparable fidelity and higher diversity.

## 5.1 Modeling High-Resolution Face Images

To further assess the effectiveness of our multi-scale approach for capturing extremely long range dependencies in the data, we train a three level hierarchical model over the FFHQ dataset [15] at $1024 \times 1024$ resolution. This dataset consists of 70000 high-quality human portraits with a considerable diversity in gender, skin colour, age, poses and attires. Although modeling faces is generally considered less difficult compared to ImageNet, at such a high resolution there are also unique modeling challenges that can probe generative models in interesting ways. For example, the symmetries that exist in faces require models capable of capturing long range dependencies: a model with restricted receptive field may choose plausible colours for each eye separately, but can miss the strong correlation between the two eyes that lie several hundred pixels apart from one another, yielding samples with mismatching eye colours.

## 5.2 Quantitative Evaluation

In this section, we report the results of our quantitative evaluations based on several metrics aiming to measure the quality as well as diversity of our samples.

### 5.2.1 Negative Log-Likelihood and Reconstruction Error

One of the chief motivations to use likelihood based generative models is that negative log likelihood (NLL) on the test and training sets give an objective measure for generalization and allow us to monitor for over-fitting. We emphasize that other commonly used performance metrics such as FID and Inception Score completely ignore the issue of generalization; a model that simply memorizes the training data can obtain a perfect score on these metrics. The same issue also applies to some recently proposed metrics such as Precision-Recall [32, 19] and Classification Accuracy Scores [28]. These sample-based metrics only provide a proxy for the quality and diversity of samples, but are oblivious to generalization to *held-out* images. Note that the NLL values for our top and bottom priors, reported in Fig. 1, are close for training and validation, indicating that neither of these networks overfit. We note that these NLL values are only comparable between prior models that use the same pretrained VQ-VAE encoder and decoder.

### 5.2.2 Precision - Recall Metric

Precision and Recall metrics are proposed as an alternative to FID and Inception score for evaluating the performance of GANs [32, 19]. These metrics aim to explicitly quantify the trade off between coverage (recall) and quality (precision). We compare samples from our model to those obtained from BigGAN- deep using the improved version of precision-recall with the same procedure outlined in [19] for all 1000 classes in ImageNet. Fig. 7b shows the Precision-Recall results for VQ-VAE and BigGan with the classifier based rejection sampling ('critic', see section 3.3) for various rejection rates and the BigGan-deep results for different levels of truncation. VQ-VAE results in slightly lower levels of precision, but higher values for recall.

`https://tfhub.dev/deepmind/biggan-deep-256/1`

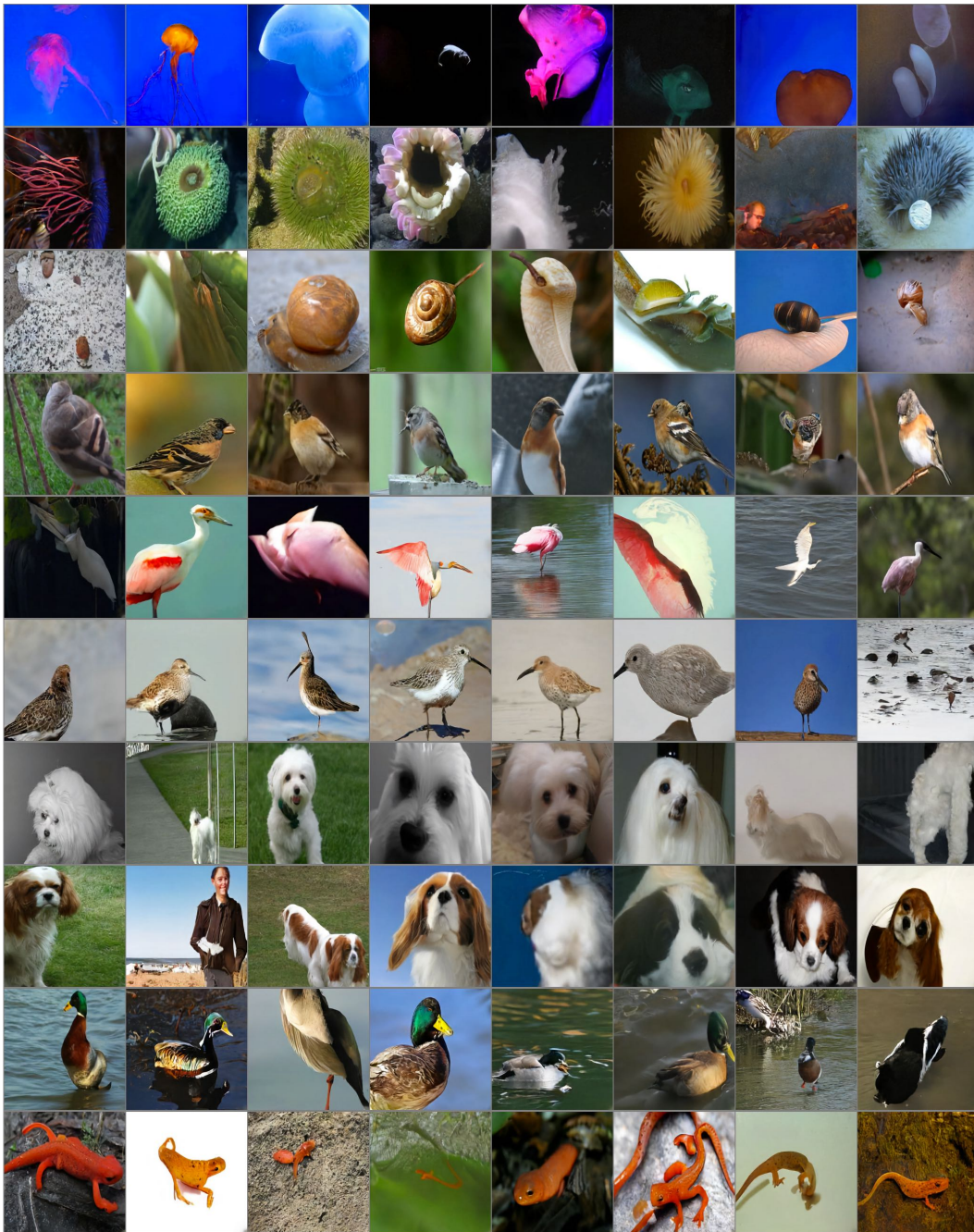

Figure 4: Class conditional random samples. Classes from the top row are: 108 sea anemone, 109 brain coral, 114 slug, 11 goldfinch, 130 flamingo, 141 redshank, 154 Pekinese, 157 papillon, 97 drake, and 28 spotted salamander.

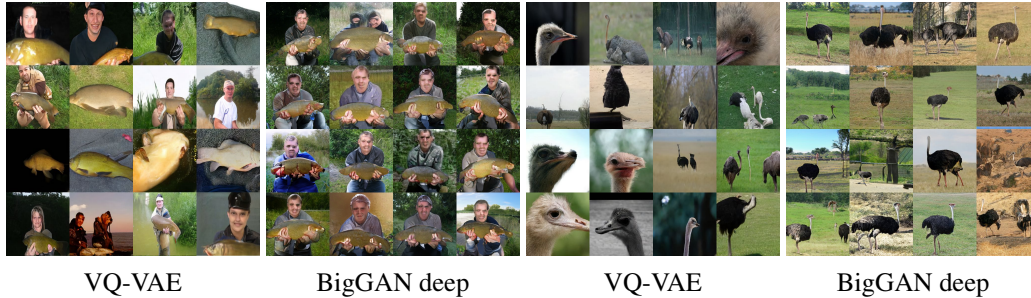

| VQ-VAE | BigGAN deep | VQ-VAE | BigGAN deep |

Figure 5: Sample diversity comparison between VQ-VAE-2 and BigGan Deep for Tinca-Tinca (1st ImageNet class) and Ostrich (10th ImageNet class). BigGAN was sampled with 1.0 truncation to yield maximum diversity. Several kinds of samples such as top view of the fish or different kinds of poses (eg, close up ostrich) are absent from BigGAN's samples. Please zoom into the pdf for inspecting the details and refer to the Supplementary material for comparison on more classes.

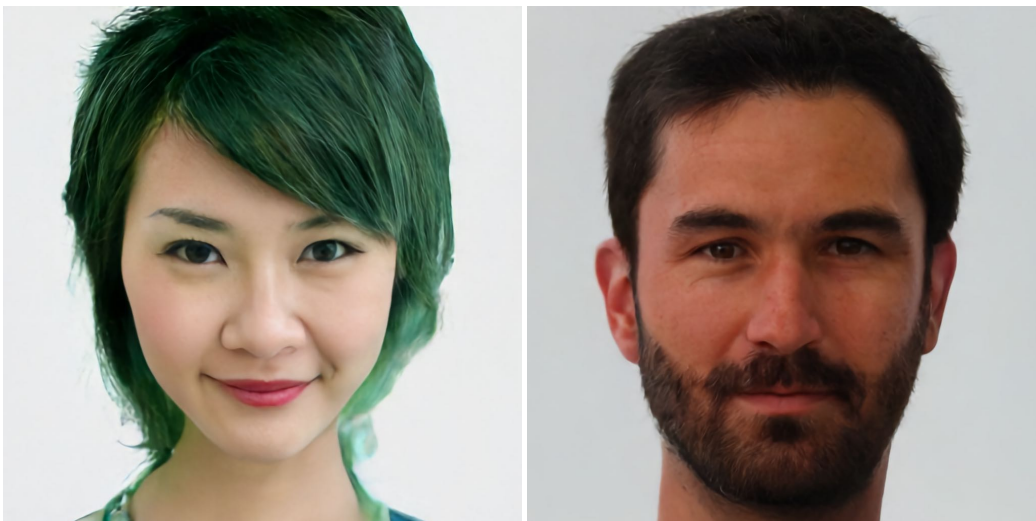

Figure 6: Representative samples from the 3-level hierarchical model trained on FFHQ-1024 × 1024. Samples capture long-range dependencies such as matching eye colour or symmetric facial features, while covering lower density data distribution modes such as green hair. See the supplementary material for more samples, including full resolution samples.

## 5.3 Classification Accuracy Score

We also evaluate our method using the recently proposed Classification Accuracy Score (CAS) [28], which requires training an ImageNet classifier only on samples from the candidate model, but then evaluates its classification accuracy on real images from the test set, thus measuring sample quality and diversity. The result of our evaluation with this metric are reported in Table 2. In the case of VQ-VAE, the ImageNet classifier is only trained on samples, which lack high frequency signal, noise, etc. (due to compression). Evaluating the classifier on VQ-VAE reconstructions of the test images closes the "domain gap" and improves the CAS score without need for retraining the classifier.

|  | Top-1 Accuracy | Top-5 Accuracy |
|---|---|---|
| BigGAN deep | 42.65 | 65.92 |
| VQ-VAE | 54.83 | 77.59 |
| VQ-VAE after reconstructing | 58.74 | 80.98 |
| Real data | 73.09 | 91.47 |

Table 2: Classification Accuracy Score (CAS) [28] for the real dataset, BigGAN-deep and our model.

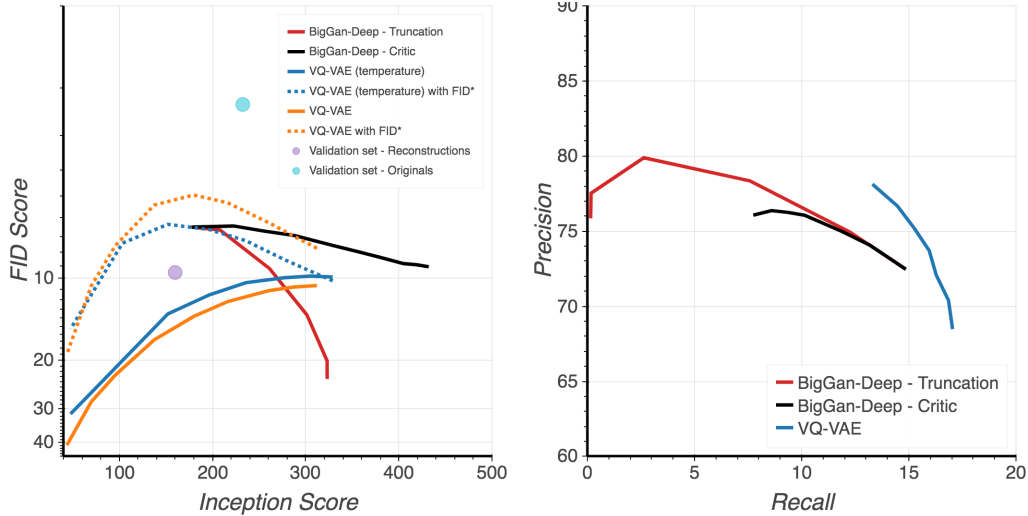

(a) Inception Scores [34] (IS) and Fréchet Inception Distance scores (FID) [13].

(b) Precision - Recall metrics [32, 19].

Figure 7: Quantitative Evaluation of Diversity-Quality trade-off with FID/IS and Precision/Recall.

### 5.3.1 FID and Inception Score

The two most common metrics for comparing GANs are Inception Score [34] and Fréchet Inception Distance (FID) [13]. Noting that there are several known drawbacks to these metrics [3, 32, 19], we report our results in Fig. 7a. We use the classifier-based rejection sampling for trading off diversity with quality (Section 3.3). For VQ-VAE this improves both IS and FID scores, with the FID going from roughly $\sim 30$ to $\sim 10$. For BigGan-deep the rejection sampling (referred to as critic) works better than the truncation method proposed in the BigGAN paper [5]. We observe that the inception classifier is quite sensitive to event slightest blurriness or other perturbations introduced in the VQ-VAE reconstructions, as shown by an FID $\sim 10$ instead of $\sim 2$ when simply compressing the originals. We therefore also compute the FID between VQ-VAE samples and the reconstructions (which we denote as FID*) showing that the inception network statistics are much closer to real images data than what the FID would otherwise suggest.

## 6   Conclusion

We propose a simple method for generating diverse high resolution images using VQ-VAE with a powerful autoregressive model as prior. Our encoder and decoder architectures are kept simple and light-weight as in the original VQ-VAE, with the only difference that we use a hierarchical multi-scale latent maps for increased resolution. The fidelity of our best class conditional samples are competitive with the state of the art Generative Adversarial Networks, with broader diversity in several classes, contrasting our method against the known limitations of GANs. Still, concrete measures of sample quality and diversity are in their infancy, and visual inspection is still necessary. Lastly, we believe our experiments vindicate autoregressive modeling in the latent space as a simple and effective objective for learning large scale generative models.

### Acknowledgments

We would like to thank Suman Ravuri, Jeff Donahue, Sander Dieleman, Jeffrey Defauw, Danilo J. Rezende, Karen Simonyan and Andy Brock for their help and feedback.

## Footnotes

[2] https://github.com/deepmind/sonnet/blob/master/sonnet/python/modules/nets/vqvae.py

[3] https://github.com/deepmind/sonnet/blob/master/sonnet/examples/vqvae_example.ipynb

[4]Samples are taken from BigGAN's colab notebook in TensorFlow hub:

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
