[Supplementary Material · VQVAE_2_Supplementary_Neurips_Camera_Ready.pdf]

# Generating Diverse High-Fidelity Images
# with VQ-VAE-2
# (Supplementary Material)

**Ali Razavi**$^*$
DeepMind
alirazavi@google.com

**Aäron van den Oord**$^*$
DeepMind
avdnoord@google.com

**Oriol Vinyals**
DeepMind
vinyals@google.com

## A   Architecture Details and Hyperparameters

### A.1   Overall Training and Sampling Procedures

The overall procedure for training the multiple stages of VQ-VAE-2 is described in Algorithm 1. We apply random cropping and flipping augmentations for Imagenet. For FFHQ, we additionally perturb brightness with maximum delta of $0.08$, random hue with maximum delta of $0.02$, and add or subtract $10\%$ random saturation and contrast. We also, with $50\%$ probability, add a Gaussian noise with standard deviation of $0.02$ to the whole image. Note that the training of the top and bottom priors can be done in parallel as the bottom prior is conditioned on the top-level discrete codes of the ground truth images in the dataset, which are obtained by running the pretrained encoder network. We used Tensorflow v1.4 [1] to implement our models. We trained all models using Google Cloud TPUv3 (https://cloud.google.com/tpu). For VQ-VAE encoders and decoders of both FFHQ and Imagenet we used 8 TPU cores, while prior networks are trained with 128 cores.

---

**Algorithm 1** VQ-VAE training (stage 1)

**Require:** Functions $E_{top}$, $E_{bottom}$, $D$, $\mathbf{x}$ (batch of training images)
1: $\mathbf{h}_{top} \leftarrow E_{top}(\mathbf{x})$

    ▷ quantize with top codebook eq 1.
2: $\mathbf{e}_{top} \leftarrow Quantize(\mathbf{h}_{top})$

3: $\mathbf{h}_{bottom} \leftarrow E_{bottom}(\mathbf{x}, \mathbf{e}_{top})$

    ▷ quantize with bottom codebook eq 1.
4: $\mathbf{e}_{bottom} \leftarrow Quantize(\mathbf{h}_{bottom})$

5: $\hat{\mathbf{x}} \leftarrow D(\mathbf{e}_{top}, \mathbf{e}_{bottom})$

    ▷ Loss according to eq 2.
6: $\theta \leftarrow Update(\mathcal{L}(\mathbf{x}, \hat{\mathbf{x}}))$

---

**Algorithm 2** Prior training (stage 2)

1: $\mathbf{T}_{top}, \mathbf{T}_{bottom} \leftarrow \emptyset$    ▷ training set
2: **for** $\mathbf{x} \in$ training set **do**
3:     $\mathbf{e}_{top} \leftarrow Quantize(E_{top}(\mathbf{x}))$
4:     $\mathbf{e}_{bottom} \leftarrow Quantize(E_{bottom}(\mathbf{x}, \mathbf{e}_{top}))$
5:     $\mathbf{T}_{top} \leftarrow \mathbf{T}_{top} \cup \mathbf{e}_{top}$
6:     $\mathbf{T}_{bottom} \leftarrow \mathbf{T}_{bottom} \cup \mathbf{e}_{bottom}$
7: **end for**
8: $p_{top} = \texttt{TrainPixelCNN}(\mathbf{T}_{top})$
9: $p_{bottom} =$ $\texttt{TrainCondPixelCNN}(\mathbf{T}_{bottom}, \mathbf{T}_{top})$

    ▷ Sampling procedure
10: **while** true **do**
11:     $\mathbf{e}_{top} \sim p_{top}$
12:     $\mathbf{e}_{bottom} \sim p_{bottom}(\mathbf{e}_{top})$
13:     $\mathbf{x} \leftarrow D(\mathbf{e}_{top}, \mathbf{e}_{bottom})$
14: **end while**

---

$^*$Equal contributions.

|  | Top-Prior ($32 \times 32$) | Bottom-Prior ($64 \times 64$) |
|---|---|---|
| Input size | $32 \times 32$ | $64 \times 64$ |
| Batch size | 1024 | 512 |
| Hidden units | 512 | 512 |
| Residual units | 2048 | 1024 |
| Layers | 20 | 20 |
| Attention layers | 4 | 0 |
| Attention heads | 8 | - |
| Conv Filter size | 5 | 5 |
| Dropout | 0.1 | 0.1 |
| Output stack layers | 20 | - |
| Conditioning stack residual blocks | - | 20 |
| Training steps | 1600000 | 754000 |
| Polyak EMA decay | 0.9999 | 0.9999 |

Table 1: Hyper parameters of autoregressive prior networks used for Imagenet-256 experiments.

## A.2 PixelCNN Prior Networks

Our top-level prior network models $32 \times 32$ latent variables. The residual gated convolution layers of PixelCNN are interspersed with causal multi-headed attention every five layers. The details of the architecture we used for the Gated layers are depicted in fig:pixelsnail-arch. To regularize the model, we incorporate dropout after each residual block as well as dropout on the logits of each attention matrix. We found that adding deep residual networks consisting of $1 \times 1$ convolutions on top of the PixelCNN stack further improves likelihood without slowing down training or increasing memory footprint too much.

Our bottom-level conditional prior operates on latents with $64 \times 64$ spatial dimensions. This is significantly more expensive in terms of required memory and computation cost. As argued before, the information encoded in this level of the hierarchy mostly corresponds to local features, which do not require large receptive fields as they are conditioned on the top-level prior. Therefore, we use a less powerful network with no attention layers. We also find that using a deep residual conditioning stack significantly helps at this level. The pseudo-code in the following provides the details of the residual conditioning stack we use in the bottom prior.

**Pseudo Code for Conditioning Stack**

```
1  def resnet_block(h, num_hiddens, num_output_hiddens):
2    h = BatchNorm(h)
3    h = Relu(h)
4    h = Conv2D(num_hiddens, kernel_shape=[1, 1])(h)
5    h = Relu(h)
6    h = Conv2D(num_hiddens, kernel_shape=[3, 3])(h)
7    h = Relu(h)
8    h = Conv2D(num_output_hiddens, kernel_shape=[1, 1])(h)
9    return h

11 def conditioning_stack(h, num_conditioning_blocks):
12   h = Conv2D(1024, kernel_shape=[1, 1], stride=[1, 1])(h)
13   for _ in range(num_conditioning_blocks // 2):
14     h += resnet_block(h, 256, 1024)
15   h2 = Conv2D(2048, kernel_shape=[1, 1], stride=[2, 2])(h)
16   for _ in range(num_conditioning_blocks // 2):
17     h2 += resnet_block(h2, 512, 2048)
18   h3 = Conv2D(2048, kernel_shape=[1, 1], stride=[2, 2])(h2)
19   for _ in range(num_conditioning_blocks):
20     h3 += resnet_block(h3, 512, 2048)
21   h2 += Conv2DTranspose(2048, kernel_shape=(2, 2), stride=(2, 2))(h3)
22   for _ in range(num_conditioning_blocks // 2):
23     h2 += resnet_block(h2, 512, 2048)
24   h += Conv2DTranspose(1024, kernel_shape=(2, 2), stride=(2, 2))(h2)
25   for _ in range(num_conditioning_blocks // 2):
26     h += resnet_block(h, 256, 1024)

28   return h
```

Figure 1: Architecture of PixelCNN prior.

|                                   | Top-Prior      | Mid-Prior      | Bottom-Prior       |
|-----------------------------------|----------------|----------------|--------------------|
| Input Size                        | $32 \times 32$ | $64 \times 64$ | $128 \times 128$   |
| Batch size                        | 1024           | 512            | 256                |
| hidden units                      | 512            | 512            | 512                |
| residual units                    | 2048           | 1024           | 1024               |
| layers                            | 20             | 20             | 10                 |
| Attention layers                  | 4              | 1              | 0                  |
| Attention heads                   | 8              | -              |                    |
| Conv Filter size                  | 5              | 5              | 5                  |
| Dropout                           | 0.5            | 0.3            | 0.25               |
| Output stack layers               | 0              | 0              | 0                  |
| Conditioning stack residual blocks| -              | 8              | 8                  |
| Training steps                    | 237000         | 57400          | 270000             |
| Polyak EMA decay                  | 0.9999         | 0.9999         | 0.9999             |

Table 2: Hyper parameters of autoregressive prior networks used for FFHQ-1024 experiments.

We optimize all models with the Adam optimizer, and use a learning rate schedule with linear warm-up and square root decay according to the following formulae:

$$LR_{imagenet-top,bottom} = 0.18 \times h_d^{-0.5} \min(step\_num^{-0.35}, step\_num \times 16000^{-1.5})$$

$$LR_{FFHQ-top,mid} = 0.324 \times h_d^{-0.5} \min(step\_num^{-0.35}, step\_num \times 16000^{-1.5})$$

$$LR_{FFHQ-bottom} = 0.18 \times h_d^{-0.5} \min(step\_num^{-0.35}, step\_num \times 16000^{-1.5})$$

Where $h_d$ is the number of hidden units specified in Table 1 for ImageNet and in Table 2 for FFHQ.

## A.3 VQ-VAE Encoder and Decoder

The details of hyper-parameters used for training the hierarchical encoder and decoder networks of Imagenet and FFHQ experiments are reported in Table 3. All models are trained with the Adam optimzer.

| | ImageNet | FFHQ |
|---|---|---|
| Input size | $256 \times 256$ | $1024 \times 1024$ |
| Latent layers | $32 \times 32, 64 \times 64$ | $32 \times 32, 64 \times 64, 128 \times 128$ |
| $\beta$ (commitment loss coefficient) | 0.25 | 0.25 |
| Batch size | 128 | 128 |
| Hidden units | 128 | 128 |
| Residual units | 64 | 64 |
| Layers | 2 | 2 |
| Codebook size | 512 | 512 |
| Codebook dimension | 64 | 64 |
| Encoder conv filter size | 3 | 3 |
| Upsampling conv filter size | 4 | 4 |
| Training steps | 2207444 | 304741 |
| Optimizer | Adam | Adam |
| Polyak EMA decay | 0.9999 | 0.9999 |
| Learning Rate | 0.0002 | 0.0002 |

Table 3: Hyper parameters of VQ-VAE encoder and decoder used for ImageNet-256 and FFHQ-1024 experiments.

## B   Ablation Studies

We studied the effect of model size, batch size and the gains of using self-attention in our prior architecture. The details of these ablations are reported in Table 4. We find that performance is affected most by the number of hidden units in the model, followed by self-attention and finally batch size. Though batch size appears not be decisive for the final performance of smaller models, using a relatively large batch-size, when possible, linearly speeds up training time.

| Batch Size NLL@1M steps | Hidden Units | Residual Size | Num Attention | NLL @64 epocchs |
|---|---|---|---|---|
| 64 3.62 | 64 | 256 | 0 | 3.62 |
| 64 3.60 | 64 | 256 | 2 | 3.60 |
| 64 3.56 | 128 | 512 | 0 | 3.56 |
| 64 3.54 | 128 | 512 | 2 | 3.54 |
| 64 3.51 | 256 | 1024 | 0 | 3.51 |
| 64 3.49 | 256 | 1024 | 2 | 3.49 |
| 128 3.62 | 64 | 256 | 0 | 3.62 |
| 128 3.60 | 64 | 256 | 2 | 3.60 |
| 128 3.55 | 128 | 512 | 0 | 3.56 |
| 128 3.53 | 128 | 512 | 2 | 3.54 |
| 128 3.50 | 256 | 1024 | 0 | 3.51 |
| 128 3.48 | 256 | 1024 | 2 | 3.49 |
| 256 3.61 | 64 | 256 | 0 | 3.63 |
| 256 3.59 | 64 | 256 | 2 | 3.61 |
| 256 3.55 | 128 | 512 | 0 | 3.57 |
| 256 3.53 | 128 | 512 | 2 | 3.55 |
| 256 3.49 | 256 | 1024 | 0 | 3.51 |
| 256 3.47 | 256 | 1024 | 2 | 3.50 |
| 1024 **3.42** | 512 | 2048 | 4 | **3.49** |

Table 4: Ablation of top prior performance with respect to batch-size, number of hidden units and attention. The rest of hyperparameters are the same as those listed in Table 1. All models are trained on the same top level VQ-VAE latent space of $256 \times 256$ Imagenet that is used in the main experiments of the paper.

### B.1   Ablation of Vector Quantization

In this section we report on our experiments to study the role of the VQ-VAE algorithm in enabling high resolution, realistic samples. As described in Section 3, our method leverages VQ-VAE as a

lossy compression information bottleneck to map the high-dimensional, high-entropy input image to a smaller, compressed latent space where much of the imperceptible information in the image is removed. Unlike the original pixel space, the density of this compressed, discrete latent space is amenable to effective modeling with autoregressive PixelCNN architectures. To demonstrate the necessity of the information-bottleneck stage, we replace VQ-VAE with unregularized autoencoders of various dimensions, and use the same prior architecture to estimate the density of the resulting latent spaces of these autoencoders. For simplicity and closer control, we run these experiments on $128 \times 128$ Imagenet, and use a single latent layer with spatial dimensionality of $32 \times 32$. The employed PixelCNN architecture has 256 hidden units and two attention layers, and is trained with batch-size of 256. Similar to [5], we used a discretized mixture of logistics distributions with 10 components and causal linear dependencies on the output dimensions to estimate the likelihood of continuous latent vectors of the unregularized autoencoders. The results of these experiments are reported in Table 5. As expected, the autoencoders are able to achieve very low distortion even with small number of latent dimensions, but at the cost of an extremely poor density estimation performance of the latent space. Unlike the VQ space, the latent spaces resulting from these unregularized autoencoders have high entropy and are overly complex to be modeled effectively with the PixelCNN priors networksw.

| Latent Dimension | Distortion PSNR (dB) | NLL (bits per Image) |
|---|---|---|
| **Unregularized AutoEncoders** | | |
| 8 | 32.41 | 36933 |
| 16 | 37.33 | 62047 |
| 32 | 45.92 | 115231 |
| 64 | 49.80 | 196484 |
| **VQ-VAE** 512 codes, 64 dimensions | 27.19 | 4919 |

Table 5: Ablation of the necessity of lossy compression.

# C   Nearest Neighbour Training Examples

In this section we report the nearest neighbours of a number of samples obtained from our Imagenet model in the training subset of Imagenet. Fig. 2 presents nearest neighbours computed in the feature space obtained from a pretrained VGG-16 [6] Imagenet classifier. We use activations of the second fully connected layer in the network. As can be seen in Fig. 2, VGG is remarkably effective in retrieving perceptually and semantically close examples. Nevertheless, while the samples share semantic properties with the nearby training examples, they are all distinctly novel. For the sake of completeness, we also demonstrate the closest neighbours based on the Euclidean distance computed in the pixel space in Fig. 3. Note that the poor performance of this method for retrieval is well understood (see for example [7]) as is evident in comparison to the VGG neighbours in Fig. 2.

Figure 2: Nearest Neighbours in the VGG feature space ordered by ascending distance to the VQVAE-2 sample on the first column.

Figure 3: Nearest Neighbours in Pixel space ordered by ascending distance to the VQ-VAE2 sample on the first column.

## D   On sample interpolations

Many generative models – such as Variational Autoencoders, Generative Adversarial Networks and Flow-based models – allow for interpolation between a set of generated samples. This is usually done by first interpolating, linearly or otherwise, between the latent vectors and then generating the image from the interpolated latents.

For VQ-VAE2, there exists no equivalent or similarly straightforward way to interpolate between samples. One problem is that naively interpolating two quantized latent vectors in general may result in a latent vector that resides outside of the codebook. The second problem is that the PixelCNN prior plays a crucial role in generating good samples in the latent space (contrary to sampling from the uniform prior in the VQ space) and there is no commonly known way of interpolating between PixelCNN samples over discrete inputs.

## E   On log-likelihoods in the image domain

VQ-VAE2 is inspired by lossy-compression where performance is usually characterized with rate-distortion curves. The VQ encoder and decoder minimize the mean square error (MSE) reconstruction

cost as the distortion metric, while a PixelCNNs (a likelihood based method) is applied in the compressed lossy space. This lossy compression stage relieves the prior networks from modeling the imperceptible details in images. Optimizing for rate-distortion rather than rate (log-likelihood) brings further benefits like faster sampling and training speed as well as global coherence in samples (due to the fact that prior network capacity is not wasted on modeling invisible artifacts). As such, we follow the established convention in the lossy compression literature and report the distortion in MSE and log-likelihood (rate) in Table 1 in the main text. Note that VQ-VAE retains the crucial advantage of likelihood based methods: a clear objective to compare models, track progress and measure overfitting and mode coverage (the properties that result in diverse samples). However, using rate-distortion optimization also entails foregoing strong log-likelihood in the image domain. This design choice yields models with inferior log-likelihood in the pixel domain precisely because the model will be punished for discarding imperceptible details in the reconstruction, the very reason that the resulting latent space is tractable for for density estimation. In practical terms, a possible way of getting log-likelihoods would be to train another PixelCNN as part of the decoder of the VQ-VAE. This, however, would defeat the purpose of the proposed method. There is also no reason to believe that this combination of autoregressive decoders conditioned on vector quantized latents would result in better log-likelihood performance than training a fully visible PixelCNN variant directly on pixels without using latent variables. To summarize, we argue that the proper model comparison methodology in this paradigm is to compare the rate-distortion curves of models on regions of interest or desired fixed operating points.

## F  On Sampling Speed

One advantage that GANs and flow models have over autoregressive models is their considerably faster sampling times. In these models, all the dimensions of the sample are computed in parallel in a single forward pass, whereas for autoregressive models, sampling in principle needs as many forward passes through the network as the number of dimensions. Concretely, we measured that sampling a single $256 \times 256$ image from BigGAN Deep takes about 100ms on a modern GPU. In contrast, naively sampling our autoregressive priors position-by-position takes about 4.5 minutes and 42 minutes for our $32 \times 32$-top and $64 \times 64$-bottom priors, respectively. However, this time can be reduced by an order of magnitude using incremental sampling techniques [2, 4] which cache and reuse intermediate activations of the network for each pixel. With such an implementation, total sampling time for a single image (in a batch of size 1) is reduced to about 4 minutes. This gets amortized down to about 3 minutes per image when using larger batches. Further speed improvement is possible by engineering custom kernels that better exploit the accelerator architectures as described for example in [3].

# G  Additional Samples

We here present additional samples from our model trained on ImageNet. All these samples are taken without any cherry-picking.

Figure 4: Random samples from class 22 Bald Eagle in ImageNet.

Figure 5: Random samples from class 11 Gold Finch in ImageNet.

Figure 6: Random samples from class 24 Grey Owl in ImageNet

**VQ-VAE (Proposed)**                    **BigGAN deep**

Figure 7: Sample diversity comparison for the proposed method and BigGan Deep for Tinca-Tinca (1st ImageNet class) and Ostrich (10th ImageNet class). BigGAN samples were taken with the truncation level 1.0, to yield its maximum diversity. There are several kinds of samples such as top view of the fish or different kinds of poses such as a close up ostrich absent from BigGAN's samples. Please zoom into the pdf version for more details and refer to the Supplementary material for diversity comparison on more classes.

$h_{\text{top}}$         $h_{\text{top}}, h_{\text{bottom}}$         Original

Figure 8: Reconstructions from a hierarchical VQ-VAE with two latent maps (top and bottom) trained on ImageNet. The rightmost image is the original. Each latent map adds extra detail to the reconstruction. These latent maps are approximately 192x and 48x smaller than the original image (respectively).

Figure 9: BigGan deep samples with truncation level 0.02 which trades diversity for sample quality.

Figure 10: Temperature 0.9 samples selected by our classifier rejection sampling technique from the mortar class (ImageNet class 666). Each grid is uniformly sampled from the shown percentage of top scoring samples out of 10000 total.

Figure 11: Representative samples from FFHQ-1024. Our model is able to capture the diversity present in the dataset while producing realistic high resolution samples. This can be noticed in the variety of age, gender, skin and hair colour, pose, facial expressions as well as the presence or absence of accessories and facial hair. Our model is able to capture the relatively rare cases of multiple persons in photos.

Figure 12: More representative samples from FFHQ-1024. Notice that, thanks to autoregressive modelling in the spatially compressed latent space, our model is quite effective in capturing longer-range dependencies in these high-resolution images and produce symmetric features such as matching eye colours or symmetric shapes of glasses and bone structure in faces.

Figure 13: Full resolution sample from FFHQ-1024

Figure 14: Full resolution sample from FFHQ-1024

Figure 15: Full resolution sample from FFHQ-1024

Figure 16: Full resolution sample from FFHQ-1024

Figure 17: Full resolution sample from FFHQ-1024

Figure 18: Full resolution sample from FFHQ-1024