[Reviews · NeurIPS 2019]

Reviewer 1



# Overall Comments: This is a nice paper in the sense that it makes the AE-based models produce really high-fidelity images as good as GAN-based models. In addition, the model also inherit the nice property of AE-based models that it does not suffer from the mode collapse issue. However, it seems to me that the only difference between this paper and the VQ-VAE paper is that this work introduces the hierarchical structure to learn different levels of latent representations and priors. The novelty looks a bit low. In addition, this paper didn't provide any idea about why such a design can make the generative performance better. The loss function (2) is not a reasonable objective to optimize considering the stop gradient operator. During the optimization procedure, the loss function may increase by taking a step in the gradient directions. This makes the algorithm like a hack and not elegant at all. # Questions: - Why you call this method VQ-VAE-2 since no variational method is used in the algorithm? - Is it possible that h_bottom encodes all the information for reconstruction and the algorithm ignores h_top completely? If not, why this cannot happen. In my opinion, h_bottom has size S/4xS/4xD, where S is the original image height/width and D is the number of channels. This is already big enough to remember the training data. Of course there is still a quantization step which can forbid it from remembering the input data. However, it seems to me the capacity is already big enough for h_bottom to perfectly reconstruct the input data. - Since h_bottom may have enough capacity to remember the training data, could it be that the model is just remembering the training data rather than generating really novel samples? - Can the algorithm do interpolation like may AE-based generative models can? I cannot find a trivial way to do this. Interpolation is also an important ability of AE-based models. - The input to the pixelCNN will at least have the size S/4xS/4. This will make the generation phase relative slow. Can the authors provide the comparison of time cost between this algorithm and the corresponding GAN model?

Reviewer 2



The proposed method produces impressive samples of high quality and high diversity, with a novel approach. The text is fairly clear and makes the method easy to grasp on a high level. However, the text is light on details. A detailed architecture description of the whole two-level hiararchy is missing, where the “conditional stacks” (L135, what are those?) should be explained/visualized, and how the different inputs to the decoder are merged, how the masked attention layers are implemented exactly, etc. Also, on a more minor note, Fig. 2a) does not quite align with Algorithm 1 (encoders both have aces to x, one also to etop, decoder has two inputs, no decoder from etop to hbottom), using the same terms in the figure as in the algorithm would help. Conceptually, the method seems to have some similarities to image compression literature, where the line of work leading to [A] use a hierarchical model, including an auto regressive model for latents, for *lossy* compression, and [B] proposes a hierarchical model similar to [25] for *lossless* compression. Further similarities arise a) from [A] training for MSE + NLL, motivated from a rate-distortion perspective, b) from the fact that both networks could in principle be used for sampling (albeit probably with much worse results, see Fig. 3 of [B]), and c) because NLL for some levels of their hierarchies can be calculated. Given the hierarchical nature of this work, a discussion or even quantitative comparison might be interesting. The comparison to BigGAN seems thorough and is convincing. I like the proposed ‘critic’. Refs [A] Minnen, David, et al. "Joint autoregressive and hierarchical priors for learned image compression." NeurIPS 2018. https://arxiv.org/pdf/1809.02736 [B] Mentzer, Fabian, et al. "Practical full resolution learned lossless image compression." CVPR 2019. https://arxiv.org/pdf/1811.12817.pdf Minor details - Fig. 2 and Fig. 5 miss a label. - Algorithm 3 mentioned on L138 is nowhere to be found.

Reviewer 3



Model Summary: The model proposed is based on the VQ-VAE model. The VQ-VAE model is a variational auto-encoder model, with quantized latent variables, and an auto-regressive prior trained to model these latent variables. It is extended using top-down hierarchical latent variables. The second contribution of the paper is to trade-off variability against quality of images by using the confidence of a classifier to reject bad samples. The paper is well and clearly written. The state of the art and related work is well covered, and good intuitions are given about the model. Conceptually, the novelty of the proposed approach is limited. The only difference to the VQ-VAE model seems to be the addition of hierarchical top down sampling, which is a standard construction in the VAE literature. On the other hand, the results obtained are excellent in terms of image quality. So the value of the paper is mostly experimental as it scales an existing model further. However, the paper does not put much focus on ablations studies (what is important to scale things up? e.x what's the impact of batch size etc..). Results: The samples obtained are impressive especially at high resolutions on FFHQ. They demonstrate that a likelihood model with sufficient capacity can generate compelling photo-realistic images. In particular, the precision-recall curves shown in Figure 5 convincingly show that the model obtains a quality of image close to that of BigGAN, while having better diversity. This is significant: models trained by maximum-likelihood, unlike GANs, are unlikely to drop parts of the training support ('mode-dropping') which makes it harder to produce compelling samples. However, this paper is not the first to achieve that, so better ablations that clearly show why it works may be desirable. In particular, the size of the model and batch sizes used are presumably significant, and a big part of why this works. If this is what it takes to make maximum likelihood work, it is better to make it evident. Therefore, ablations on Model size could be desirable. How does this model perform compared to existing models when model sizes are comparable? In this respect, the authors have provided architectural details in the Supplementary. Similarly, I would be interested to know how this model performs without quantization. No BPD measurements (or bounds) are provided in the main paper. I find this choice a little surprising. Given that the model belongs to the family of maximum-likelihood models, it could be desirable to report these values and compare to that literature as well as BigGAN. Classifier based rejection sampling: This contribution is quite orthogonal. In practice, resampling favours model that over-generalise VS models that mode-drop: the model is no longer penalized for generating bad samples, but has better support. In particular, given enough rejections any model will eventually produce a compelling sample. But it is indeed a simple and nice way to trade-off variability for quality, and to obtain precision recall curves. In the case where the classifier is trained on an other dataset, this uses extra data. Could you elaborate on the range of thresholds used for classifier based rejection in Figure 5? Are they the same for both VQ-VAE and BigGan? Also, is classifier based rejection used in Table 1? Minor remarks: In terms of evaluating image quality, showing nearest neighbours in pixel space and vgg-feature space could be considered.

[Author Response · NeurIPS 2019]

**Author Response for "Generating Diverse High-Fidelity Images with VQVAE-2"**

We thank the reviewers for the detailed and constructive feedback. All reviewers were impressed by the quality of our

samples. R2 had positive remarks about the significance of our method and the thoroughness of our evaluation. R3 was

satisfied with the clarity of the writing. There were however some concerns about additional results, architecture details

and novelty of the approach. We hope that we have addressed the requested clarifications below, explaining how each

will be improved in the final paper. We believe these clarifications will resolve all reviewers' concerns, but would be

happy to consider any additional suggestions.

**General response regarding to:**

**R1 -** *Name of the method*: The name VQ-VAE was coined by previous authors, so we decided to adopt the same name,

rather than using a new one. Moreover, it is possible to frame VQ-VAE in a variational framework as well (with a delta

posterior and a uniform prior), which is discussed in the original VQ-VAE paper.

**R1 -** *h_bottom encoding everything*: **A** h_bottom is preceded by an Information-Bottleneck (similar to the KL of a

regular VAE) so every latent can at most encode $\log_2(N)$ bits, where N is the size of the VQ codebook. In our case, this

is 9bits per latent. There is one latent for every 16 pixels, each having 3 color channels with 8 bits each, so this yields

a compression factor of 4*4*3*8/9=42.67. **B** Adding h_top results in much better reconstruction MSE and sharper

reconstructions. We also visualize reconstruction from h_top only (Fig. 3 in the paper and Fig. 5 in the appendix),

showing that h_top has indeed encoded quite a lot of information.

**R1 -** *Model remembering data*: The VQ-VAE is able to reconstruct test set images equally well as training images

(MSE, and qualitatively), which demonstrates that it generalizes to unseen data. Similarly, for the PixelCNN, the NLLs

of test data are comparable to those of training data.

**R1 -** *Interpolations*: There is indeed no simple way to do interpolations, which We will clarify in the final version.

**R1 -** *Speed*: Generation is slower than GANs, but faster than other autoregressive approaches that model images in the

pixel space (about 45x faster). We also implemented incremental sampling (as in Paine et al. arxiv.org/abs/1611.09482)

to cache intermediate activations that can considerably reduce sampling time. We will add a comparison in the final

version.

**R1 -** *Objective with stop-gradient not elegant*: As noted in the paper, in equation 3, we use the Exponential Moving

Average version which does not use stop-gradients. The loss in equation 2 is included for sake of completeness. These

are both different neural implementations of the K-means algorithm, which has a long established track record in many

areas of machine learning. We would also like to point out that elegance is a subjective matter, and simplicity is a form

of elegance we strove for in this work: indeed, VQ-VAE is quite simple and can be implemented in just a few lines of

code.

**R2 -** *Detailed architecture*: We agree that the architecture description could be more detailed. We will make sure that

our architecture is thoroughly specified in our final version and we will include all details and hyperparameters.

**R2 -** We will fix minor details, also cite [A], [B] to emphasize the connection with lossy-compression.

**R3 -** *Novelty / This paper is not the first to achieve that*: We are not aware of any prior works that show comparable

sample quality to BigGAN (which is SOTA) while having better diversity in any model class (let alone among

likelihood-based methods). For faces, the best prior works (ie, Glow and SPN) used a much simpler dataset CelebaHQ

with 256x256 resolution. Still their their samples look less realistic and have lower fidelity than our 1024x1024 samples

from more complex FFHQ. The only other model to achieve this has been StyleGAN, which also has the discussed

diversity limitations of GANs.

**R3 -** *Ablations wrt. model size*: We will add ablations of our model wrt. model size and batch size, but the result is that

larger models get better results. Comparison with other works, however, shows that scaling up is necessary but not

sufficient: our model gets better results compared to models with similar (or larger) size, batch-size, and compute

requirements: Menick et al. 2018, Defauw et al. 2019. The same applies for BigGAN.

**R3 -** *No quantization*: The model does not work at all without quantization. We will add this ablation in the Appendix

(with a reference from the main text).

**R3 -** *BPD measurements*: As R2 has noted, this model is inspired by lossy-compression where performance is usually

characterized with rate-distortion curves. We apply log-likelihood based methods in a compressed lossy space, thus not

having to model imperceptible details in images. This is where benefits like speed, global coherence, etc., come from.

We do report BPD in the latent spaces. Trying to go back to the pixel-domain would defeat the main purpose of the

method. That said, if there is truly interest in this metric, it is straightforward to estimate and add it to the final version

of the paper.

**R3 -** *Classifier based rejection sampling*: All samples in the paper are without the rejection sampling except for Fig 8

and 9 in the appendix where we aim to illustrate the effect of various rejection thresholds. Similarly the numbers in

Table 1 do not use this. CBRS is only used for the P/R and FID/IS curves in Fig. 5. of the main text.

**R3 -** *Nearest Neighbours*: As noted in the paper and in our response to R1, our model can be directly assessed for

overfitting by comparing train and test NLL. Nevertheless, we will include nearest neighbours in the pixel and VGG

spaces in the final version.

59

[Meta-Review · NeurIPS 2019]

This paper presents great visual images and quantitative scores for an autoencoder-based generative model. All reviewers agree on this aspect, and this is primarily the reason why acceptance is warranted. Certainly an AE pipeline with this capability is a worthwhile contribution to the community. However, the proposed method is mostly some engineered enhancements to the basic VQ-VAE model that has already been published. Moreover, full architectural details and hyperparamter settings were not provided in the original submission but were promised for the final version. For an enhanced AE model with modest novelty, this is a bit problematic and at least partially obfuscates proper evaluation of the complete system by reviewers. Even so, I would give the authors the benefit of the doubt and assume that the final version will include all the missing details.